# Need-based prioritization of behavior

C Joseph Burnett[1,2,3], Samuel C Funderburk[1,2†], Jovana Navarrete[1,2], Alexander Sabol[1,2], Jing Liang-Guallpa[1,2], Theresa M Desrochers[4], Michael J Krashes[1,2*]

[1]Diabetes, Endocrinology, and Obesity Branch, National Institute of Diabetes and Digestive and Kidney Diseases (NIDDK), National Institutes of Health, Bethesda, United States; [2]National Institute on Drug Abuse (NIDA), National Institutes of Health, Baltimore, United States; [3]Brown University Graduate Partnerships Program, Providence, United States; [4]Department of Neuroscience, Brown University, Providence, United States

**Abstract** When presented with a choice, organisms need to assimilate internal information with external stimuli and past experiences to rapidly and flexibly optimize decisions on a moment-to-moment basis. We hypothesized that increasing hunger intensity would curb expression of social behaviors such as mating or territorial aggression; we further hypothesized social interactions, reciprocally, would influence food consumption. We assessed competition between these motivations from both perspectives of mice within a resident-intruder paradigm. We found that as hunger state escalated, resident animal social interactions with either a female or male intruder decreased. Furthermore, intense hunger states, especially those evoked via AgRP photoactivation, fundamentally altered sequences of behavioral choice; effects dependent on food availibility. Additionally, female, but not male, intrusion attenuated resident mouse feeding. Lastly, we noted environmental context-dependent gating of food intake in intruding mice, suggesting a dynamic influence of context cues on the expression of feeding behaviors.
DOI: https://doi.org/10.7554/eLife.44527.001

*For correspondence:
michael.krashes@nih.gov

Present address: †Washington University School of Medicine, St. Louis, United States

Competing interests: The authors declare that no competing interests exist.

## Introduction

A motivated behavior is purposive in attaining an explicit goal, aroused through internal representations acting in concert with the presence of incentive stimuli and learned associations (*Bindra, 1968*; *Toates, 1980*; *Toates, 1981*). Due to the intricacy of these motivational systems, they are often simplified and studied in a behavioral vacuum, encouraging subjects to express goal-seeking behavior toward one objective. Yet, individual systems likely never operate in isolation from one another; instead they compete for expression. When animals harbor rival incentives and only mutually exclusive courses of action will satisfy those incentives, they are forced to make a choice (*Sherrington, 1906*). Therefore, neural computations must weigh the ramifications of these stimuli for optimal behavioral choice.

As an entry point into the control of feeding, we used Agouti-related peptide (AgRP)-expressing neurons in the arcuate nucleus (ARC) and evaluated their capacity to regulate food intake in discrete, ethologically-relevant contexts. $ARC^{AgRP}$ neurons are sensitive to a number of circulating endocrine signals (*Cowley et al., 2003*; *Takahashi and Cone, 2005*) and display appetite-state and calorie-dependent dynamic activity changes (*Betley et al., 2015*; *Beutler et al., 2017*; *Chen et al., 2015*; *Mandelblat-Cerf et al., 2015*; *Su et al., 2017*). Stimulation of these neurons before or during food accessibility drives motivated feeding (*Aponte et al., 2011*; *Chen et al., 2016*; *Krashes et al., 2011*). Conversely, inhibition of $ARC^{AgRP}$ activity directly (*Betley et al., 2015*; *Krashes et al., 2011*; *Vardy et al., 2015*) or via activation of upstream inhibitory populations (*Garfield et al., 2016*) attenuates feeding, while postnatal ablation induces starvation (*Gropp et al., 2005*; *Luquet et al.,*

*2005*). Recent studies demonstrated ARC$^{AgRP}$ neurons have the capacity to orchestrate food-seeking to the exclusion of self-preservation and social behaviors (*Burnett et al., 2016*; *Dietrich et al., 2015*; *Padilla et al., 2016*), but deeper behavioral analyses are needed to fully understand this relationship. Therefore, we investigated how varying degrees of natural and ARC$^{AgRP}$-evoked hunger impact this interface.

We discovered a complicated interplay between these motivation systems. First, we found hunger had a profound effect on a resident male's copulatory advances towards a female conspecific, but only if food was present. Similarly, when exposed to a subordinate male intruder, residents displayed less aggression in more intense hunger states; especially when food was present. However, in contrast to the reduced food intake of resident males in the presence of a female intruder compared to no intruder, resident male feeding remained intact in the presence of a male intruder compared to no intrusion. Sequence analyses of behavioral transitions revealed distinct patterns of behavior in hungry animals in the presence of food, driven primarily by caloric consumption, with the highest degree of behavioral sequence disruption observed during unnatural ARC$^{AgRP}$ stimulation. Remarkably, all states of hunger exhibited comparable patterns of behavior in the absence of food. Additionally, we found that intruding receptive females and subordinate males reduced food consumption, independent of appetite state, in an empty or occupied territorialized cage compared to food intake in their homecages. Meanwhile, extreme physiological, but not artificially-induced hunger, abrogated female receptivity to male advances while male intruders' defensive behavior remained analogous across appetite condition. Together, these findings paint a complicated picture of real-time behavioral decision-making that cannot easily be captured in more canonical assays of motivated behavior that sacrifice ecological validity for tighter experimenter control. We argue that experimental designs addressing roles of competing motivational drives more accurately reflect the need to integrate multiple, disparate streams of information from real-world stimuli to mold appropriate behavior. Thus application of such paradigms is critical for our ultimate understanding of behavior.

## Results

### Prolonged food-deprivation leads to more pronounced changes in rodent physiology

Although hunger is often considered a stable construct, whole body physiology undergoes robust alterations in response to lengthening periods of food deprivation, which likely contribute to shaping behavioral output. We began by measuring body weights and composition of male and female C57BL/6 mouse cohorts at three distinct time points: (1) *ad libitum* food access, (2) fasted for one dark cycle (18 hr food deprivation) and (3) fasted for two dark cycles (48 hr food deprivation) (*Figure 1—figure supplement 1A*). Withholding food access overnight robustly decreased body weight, fat mass and lean mass in every mouse, independent of sex; these effects were exacerbated in the 48 hr food-deprived condition (*Figure 1A–C*).

In another cohort of mice, we measured tissue weights and serum levels of key circulating factors. We again observed substantial weight loss in groups of food-deprived animals at time of dissection compared to their *ad libitum* fed counterparts (*Figure 1—figure supplement 1B*). We also found that increasing levels of food-deprivation reliably reduced weights of both inguinal and epididymal white adipose tissue, quadriceps, gastrocnemius, liver and stomach tissue (*Figure 1D*). Furthermore, fasting appreciably lowered concentrations of numerous circulating factors including glucose, triacylglycerol, insulin, leptin and lactate while boosting levels of free fatty acids, corticosterone, beta-hydroxybutyrate and liver triacylglycerol (*Figure 1—figure supplement 1C–F*). Importantly, for many of the factors assessed, the degree of modification was exacerbated in the Fasted (48 hr) versus Fasted (18 hr) condition.

### Receptive female intruders induce conflict between feeding and social interaction in resident male mice

#### Experimental design

To evaluate the role hunger plays on the innate drive to reproduce and determine the capacity of pursuit/mating in regulating feeding, we utilized a resident-female intruder paradigm. In addition to

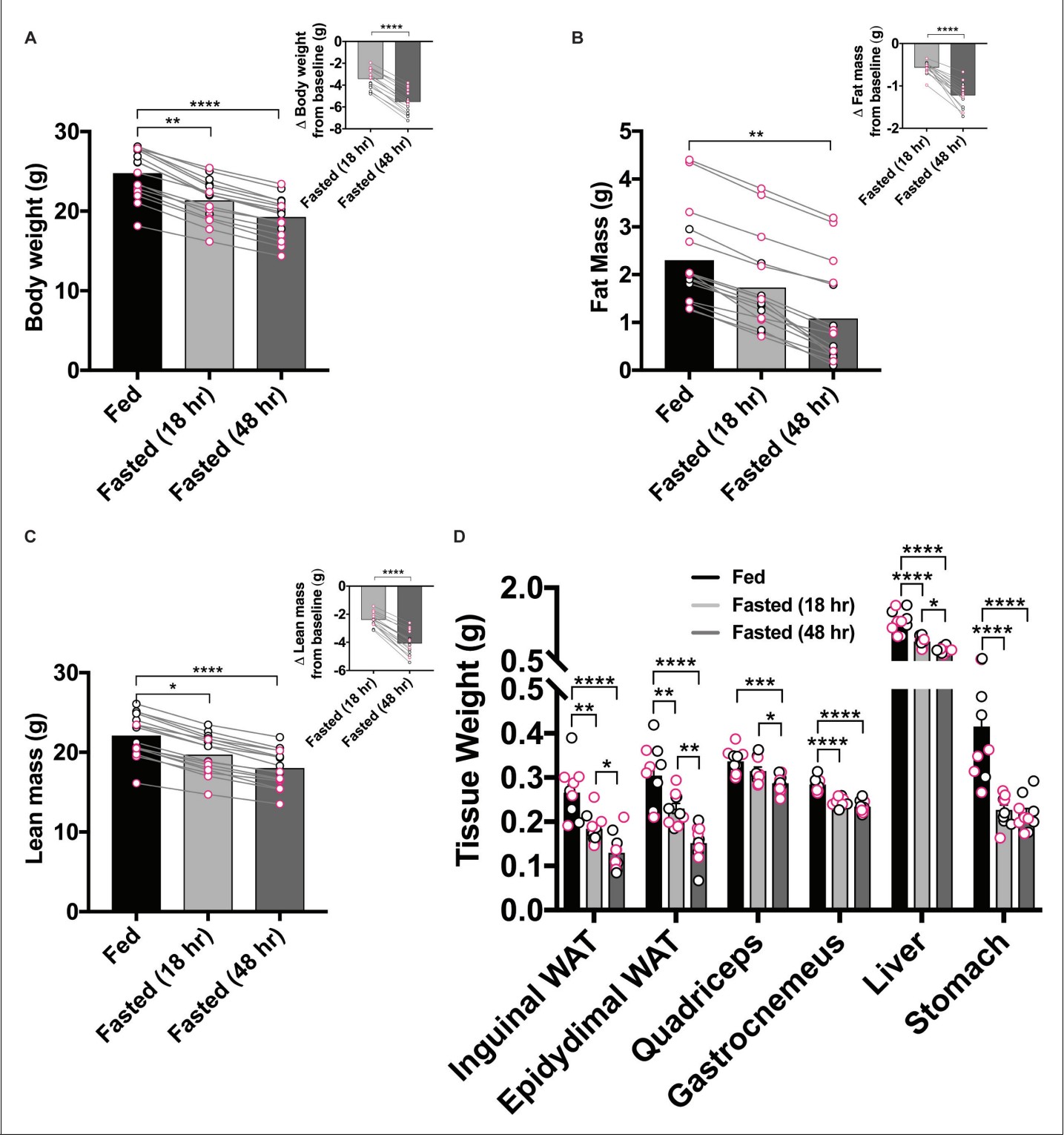

**Figure 1.** Prolonged food-deprivation leads to more pronounced changes in rodent physiology. (A–F) 18 hr food restriction reduced body weight (A), fat mass (B), lean mass (C), and tissue weights of inguinal WAT, epididymal WAT, quadriceps, gastrocnemius, liver and stomach (D) in both males and female mice, and was further accentuated following 48 hr restriction. Data points outlined in black indicate males; n = 5, Data points outlined in pink indicate females; n = 5 per group, values are means ± SEM. *p<0.05, **p<0.01, ***p<0.001.

DOI: https://doi.org/10.7554/eLife.44527.002

The following figure supplement is available for figure 1:

*Figure 1 continued on next page*

*Figure 1 continued*

**Figure supplement 1.** Prolonged food-deprivation leads to more pronounced changes in rodent physiology.
DOI: https://doi.org/10.7554/eLife.44527.003

studying physiological satiety versus hunger, we investigated artificial 'hunger' induced by selective activation of ARC$^{AgRP}$ neurons in sated animals; this ectopic activation is often used to assess ARC$^{AgRP}$ neurons' sufficiency to recapitulate behavior patterns observed during natural hunger (*Alhadeff et al., 2018*; *Aponte et al., 2011*; *Burnett et al., 2016*; *Chen et al., 2016*; *Dietrich et al., 2015*; *Goldstein et al., 2018*; *Krashes et al., 2011*; *Padilla et al., 2016*). We unilaterally targeted a Cre-dependent channelrhodopsin-2 (ChR2) virus to the ARC of *Agrp-ires-Cre* transgenic mice and implanted an optical fiber over this region to allow photostimulation (*Figure 2—figure supplement 1A*). Using this strategy, we tested subjects under five distinct appetite conditions 1) *ad lib* Fed, 2) Fasted (18 hr), 3) Fasted (48 hr), 4) *ad lib* Fed with concurrent ARC$^{AgRP}$ stimulation (Fed$^{AgRP\ CS}$) and 5) *ad lib* Fed with 20 min pre-ARC$^{AgRP}$ stimulation before experiments began (Fed$^{AgRP\ PS}$); thus, we made within-subject comparisons across conditions, a critical advantage when challenged with the known variability of social behavior (*Figure 2—figure supplement 1A*). Following surgeries, experimental animals were housed in specialized cages (*Figure 2—figure supplement 1B*).

The 60 min resident-female intruder paradigm involved three phases, each lasting 20 min. In the Pre-Phase, territorialized residents were tethered to an optical patch cord and allowed to acclimate after experimenter handling in the absence of food (*Figure 2—figure supplement 1E* left panel). During this time, *ad libitum* fed, virgin, group-housed female intruders were selected that tested positive for the estrus phase of the estrous cycle using vaginal cytology (*Figure 2—figure supplement 1C–D*). The Trial-Phase was either conducted in a) the presence of food and absence of an intruder, b) the presence of food and a female intruder, or c) the absence of food and presence of a female intruder (*Figure 2—figure supplement 1E* middle panel). During the Post-Phase, the intruder was removed from the cage to assess 'residual' resident post-trial feeding behavior (*Figure 2—figure supplement 1E* right panel).

## Behavioral choice robustly and reciprocally affects overall feeding and mating behaviors

Motivated behaviors are immensely sophisticated, but consist of several simple, stereotyped actions. To deconvolute the natural complexity of these behaviors, we focused on seven easily, reproducibly identified behavior groups in our resident animals when interacting with a female intruder in their homecage: mounting and/or intromission of the female intruder (red), pursuit and/or attempted mounting of the female intruder (yellow), anogenital chemoinvestigation (orange) and nose-to-nose chemoinvestigation (green) as interactive behaviors and drinking (blue), grooming (purple) and eating (black) as individual behaviors (*Figure 2A*, top). Each of these actions were scored on a second-by-second basis resulting in a behavioral barcode defining each resident animal's behavior over the 20 min experimental trial period (*Figure 2A*, bottom and *Figure 2—figure supplement 2A*).

In the absence of a female intruder, resident mice demonstrated need-based escalation of food consumption during the Trial Phase (*Figure 2B*). Accordingly, subjects' latency to consume 0.1 grams or five pellets, a metric used to determine homeostatic feeding (see Materials and methods), showed similar timescales across all hunger states (*Figure 2C*). When a female intruder was present, residents displayed similar grades of food consumption dependent on satiety state (*Figure 2B*): hunger provoked residents to consume more food at a faster rate than the same mice in the Fed state (*Figure 2C*). However, food intake was significantly attenuated in the presence, compared to the absence, of a female intruder, regardless of appetite condition (*Figure 2—figure supplement 2B–F*). Moreover, presence of a female intruder was sufficient to right-shift cumulative feeding curves in all conditions tested (*Figure 2C–D*; *Figure 2—figure supplement 2 G-K*). Therefore, in both physiological and non-physiologically-induced hunger states, access to a receptive female readjusts goal-seeking and blunts food intake. We found that resident food intake was significantly enhanced during the Post-Phase after the intruder female was removed compared to the same mice during this period without prior exposure to the female intruder (*Figure 2—figure supplement 3A–F*),

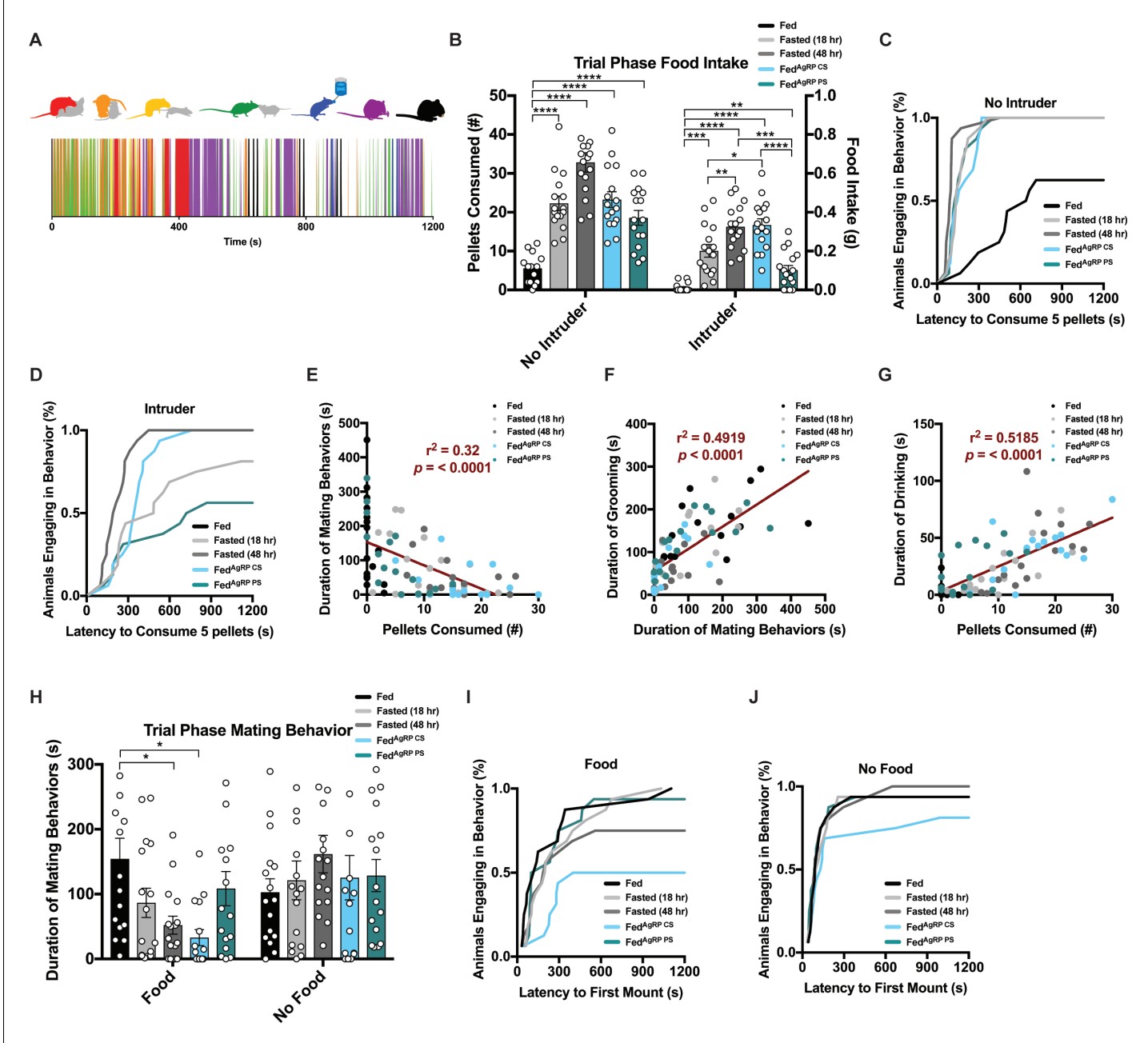

**Figure 2.** Evaluating resident behavior in response to a receptive female intruder. (**A**) Catalogue of scored resident behaviors (top); mounting and/or intromission of the female intruder (red), pursuit and/or attempted mounting of the female intruder (yellow), anogenital chemoinvestigation (orange), nose-to-nose chemoinvestigation (green), drinking (blue), grooming (purple) and eating (black). Sample raster plot of second-to-second scored behaviors across the entire Trial Phase (bottom). (**B–D**) Quantified between-subject comparisons of resident food intake (**B**) and cumulative latency plot to consume 0.1 grams of food in the absence (**C**) or presence of a female intruder (**D**). (**E–G**) Linear regression plots between mating and feeding (**E**), grooming and mating (**F**) and drinking and feeding (**G**). (**H–J**) Quantified between-subject comparisons of resident mating behavior (**H**) and cumulative latency plot to first mount in the presence (**I**) or absence of food (**J**). n = 16 per group, values are means ± SEM. *p<0.05, **p<0.01, ***p<0.001.

DOI: https://doi.org/10.7554/eLife.44527.004

The following figure supplements are available for figure 2:

**Figure supplement 1.** Experimental design of resident-intruder assay during evaluation of resident behavior.
DOI: https://doi.org/10.7554/eLife.44527.005

**Figure supplement 2.** Evaluating resident behavior in response to a receptive female intruder.
DOI: https://doi.org/10.7554/eLife.44527.006

*Figure 2 continued on next page*

*Figure 2 continued*

**Figure supplement 3.** Post-trial phase food intake in resident mice in female intruder paradigm.
DOI: https://doi.org/10.7554/eLife.44527.007

indicating the delay in the replenishment of caloric balance in the presence of a female intruder is rescued after removal of the female.

We next sought to determine whether the attenuation of resident animal feeding was the result of behaviors directed toward the female intruder. We observed that the total duration of mating behaviors, including pursuit, attempted mounting, mounting and intromission decreased with natural hunger intensity in the presence of food (*Figure 2H*). Cumulative curves to initial mounting attempt also depicted higher latencies and a lower number of total animals reaching threshold in the Fasted (48 hr) and Fed$^{AgRP\ CS}$ conditions (*Figure 2I*). Notably, the time spent engaging in mating behavior was lowest in the Fed$^{AgRP\ CS}$ state, an unnatural condition as ARC$^{AgRP}$ neurons should be silent in the presence of food.

A closer survey of each animal's behavioral barcodes (*Figure 2—figure supplement 2A*) suggested that many behaviors were interrelated. We hypothesized that diminished feeding in the presence of a female intruder resulted from direct interactions, rather than mere intruder presence. Indeed, duration of mating behavior was significantly, negatively correlated with the amount of food consumed during the 20 min Trial-Phase across grouped animals (*Figure 2E*). Grooming often follows mating behavior, particularly after ejaculation, and we observed a positive correlation between mating and grooming across grouped animals (*Figure 2F*). Consumption of dry chow drives thirst, thus water intake was strongly correlated with food consumption across grouped animals (*Figure 2G*).

## Motivational systems exhibit flexible computations contingent on accessibility to external incentives

Based on the findings that escalating hunger drives food intake and reduces mating behavior, we hypothesized that similar patterns would emerge if food was unavailable, since hungry animals may still prioritize food-seeking. However, when we conducted intruder trials where food was absent (*Figure 2—figure supplement 1E* middle panel, bottom), we found that residents showed comparable levels of time spent participating in mating behaviors, regardless of caloric need state tested (*Figure 2H*). Additionally, the latency to initially mount the female intruder was reduced and the number of animals engaging in mating behaviors increased in each condition (*Figure 2I–J*, *Figure 2—figure supplement 2Q-U*). More specifically, mice that curbed mating behavior when food was available, namely the Fasted (48 hr) and Fed$^{AgRP\ CS}$ conditions, displayed significantly more time engaging in mating behaviors when food was unavailable (*Figure 2—figure supplement 2L–P*). Thus, preventing animals from sating caloric hunger tipped behavioral choice in favor of sating reproductive drive, the only other satiable motivation at the time.

## Satiating substrate choice substantially impacts behavioral sequences in resident males exposed to female intruders

We next probed whether varying hunger levels could disrupt sequences of behaviors. To accomplish this, we deduced the likelihood that each animal, in each condition, would transition from one scored behavior into another scored behavior. Since we determined above that behavior rapidly adapts based on available choice, these analyses were calculated both in the presence of food and a female intruder (*Figure 3A*) and in the absence of food but presence of a female intruder (*Figure 3B*). Using the Fed group as a baseline, we found that certain sequences of behavior occurred with high contingency in the Fed state, including the transition from pursuit/attempted mounting into mounting/intromission, nose-to-nose chemoinvestigation into ano-genital chemoinvestigation, and mounting/intromission into grooming, while others occurred infrequently (*Figure 3A*, far left). Importantly, we found that these patterns in behavior differed in the same mice under physiological and artificial hunger. For example, we discovered the probability to transition from pursuit/attempted mounting into mounting/intromission or from mounting/intromission into

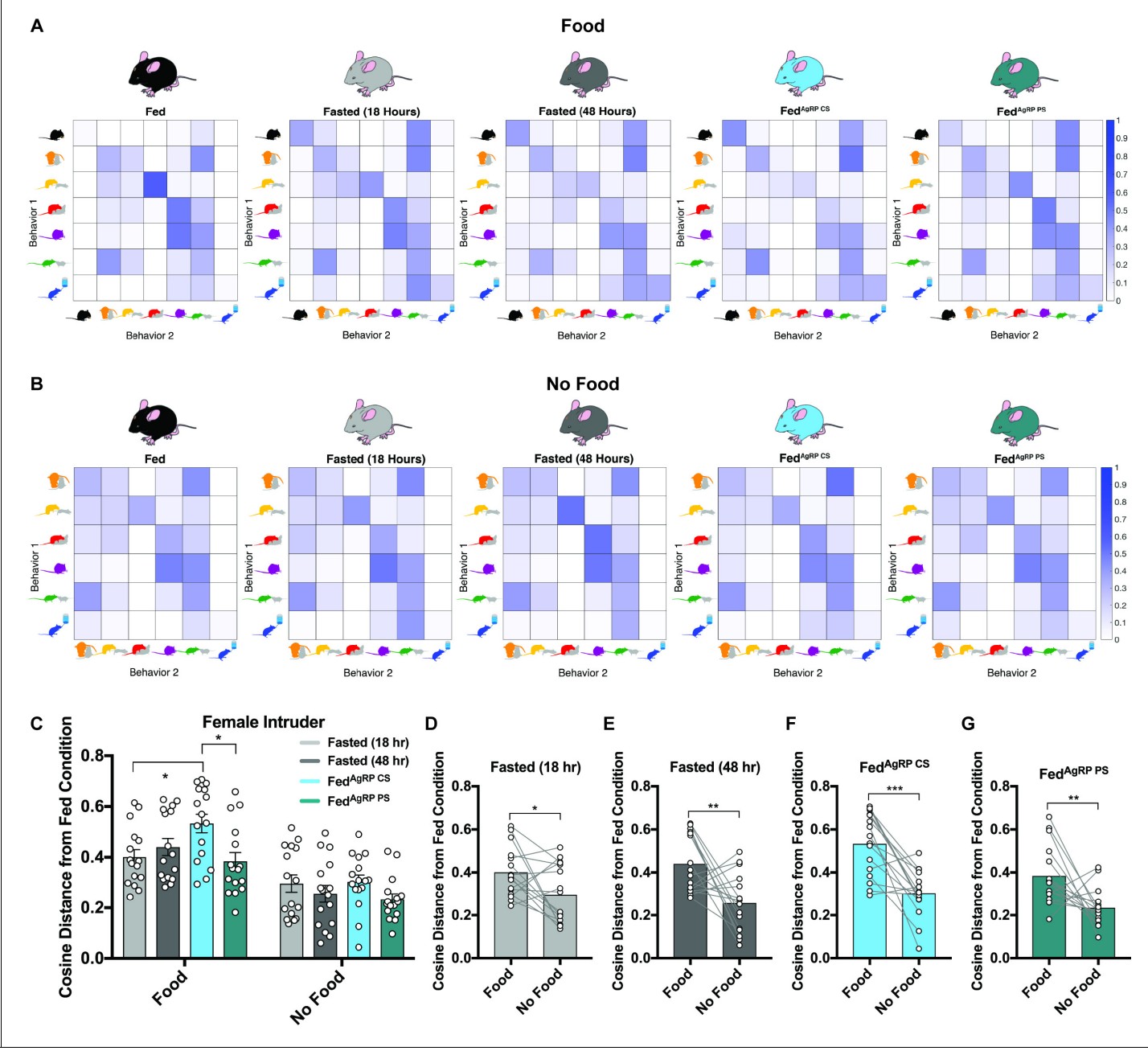

**Figure 3.** Behavioral sequence analyses of resident males exposed to a female intruder. (A–B) Transition matrices of behavioral transitions grouped by appetite condition in the presence (A) or absence (B) of food. (C) Quantified between-subject comparisons of the cosine distance from the Fed condition in the presence or absence of food, respectively. (D–G) Quantified within-subject comparisons of the cosine distance from the Fed condition in the presence versus absence of food. n = 16 per group, values are means ± SEM. *p<0.05, **p<0.01, ***p<0.001.
DOI: https://doi.org/10.7554/eLife.44527.008

grooming decreased while the probability from eating to eating or drinking to drinking increased in all hungry animals irrespective of how it was elicited (*Figure 3A*).

Within each animal, we then used the cosine distance method to compare the difference between all transition frequencies in the Fed state from the same frequencies in each natural and artificial hunger condition. This method does not rely on the frequency of observations, making it ideal for our purposes because our subjects display varying behavioral frequencies and, therefore, numbers of transitions observed. A greater cosine distance from the Fed state indicates a greater difference in

the overall patterns of behavior. During the resident-female intruder paradigm, we found that the cosine distance between transition frequencies in the Fed state displayed significant differences to all conditions of hunger, with the largest discrepancy seen in the Fed$^{AgRP\ CS}$ condition (*Figure 3C*).

We also analyzed these sequences in conditions when food was absent. To this end, we calculated average transition frequencies between the remaining six behaviors and found that transitions between behaviors across all five conditions remained fairly similar (*Figure 3B*). In agreement with this, the cosine distances from the Fed state were comparable across all natural and artificial hunger states (*Figure 3C*). Of note, all differences from the Fed state were significantly lower when food was absent compared to trials where food was present (*Figure 3D–G*).

## Receptive female intruders display context- and State-Dependent Feeding and Social Interactions with Resident Male Mice

### Female intruders display environmental context-dependent food consumption

We next analyzed how hunger state regulates a sexually mature, virgin female intruder's behavior in a resident-intruder assay. Surgeries and testing conditions were identical to protocols described above (*Figure 4—figure supplement 1A–C*). After a 20 min Pre-Phase in the absence of food, (*Figure 4—figure supplement 1D*, left panel) we took 20 min food intake measurements (Trial-Phase) of each intruder in three distinct environmental contexts: (1) the intruder's home cage, (2) a territorialized, empty male resident cage, and (3) a territorialized, occupied male resident cage (*Figure 4—figure supplement 1D*, middle panel). As predicted, we observed appetite state-dependent home cage food intake, as all hunger conditions displayed significantly elevated food consumption compared to the Fed group with rapid onset (*Figure 4B–C*). Remarkably, we found that while this state-dependent stratification of food intake remained, the total amount of food consumed was attenuated in the territorialized, empty male resident cage (*Figure 4B*; *Figure 4—figure supplement 2A, C,E,G,I*), accompanied by increased latency to consume 0.1 grams of food and reduced number of animals meeting this threshold (*Figure 4D*; *Figure 4—figure supplement 2B,D,F,H,J*). Moreover, food intake and rate of consumption were further mitigated in the territorialized, occupied male resident cage (*Figure 4B,E*; *Figure 4—figure supplement 2A–J*). Finally, when assessing food intake in the Post-Phase after the male resident had been removed from the cage (*Figure 4—figure supplement 1D*, right panel), we found a familiar pattern of feeding behavior emerge depending on the intensity of deprivation (*Figure 4—figure supplement 2K*).

## Extreme states of food deprivation alter mating behaviors

As with the resident male behavior, a repertoire of 7 female intruder behaviors were scored throughout each Trial-Phase: both interactive behaviors including receptivity, encompassing presentation/lordosis (red), non-receptivity, comprising of escape/attempted escape (yellow), anogenital chemoinvestigation (orange) and nose-to-nose chemoinvestigation (green) and individual behaviors comprising drinking (blue), grooming (purple) and eating (black) were evaluated (*Figure 4A*, top). Again, behaviors were scored each second over the entire length of the 20 min Trial-Phase and behavioral barcodes were generated for each animal (*Figure 4A*, bottom).

Although severe food deprivation is known to disrupt ovulation rate, cyclic behavior, and reproductive receptivity (*Cooper et al., 1970*; *Howland, 1971*), we still confirmed all female intruders were in estrus before the resident female-intruder assay (*Figure 4—figure supplement 1C*). We observed a clear reduction in the duration of receptivity behavior and total number of subjects engaging in receptive behaviors in Fasted (48 hr) mice that differed from all other groups tested (*Figure 4F–G*), while the duration of time spent engaging in non-receptive (escape/attempted escape) behaviors was comparable across conditions (data not shown).

## Subordinate male intruders fail to influence feeding in resident mice

### Experimental design

Next, we examined the integration of and potential conflict arising between feeding and defending a designated territory using a slightly modified resident male-intruder assay (*Koolhaas et al., 2013*). A separate cohort of male residents was prepared as described above (*Figure 2—figure supplement 1A–B*), and these sexually-experienced animals were provided with subordinate mice to

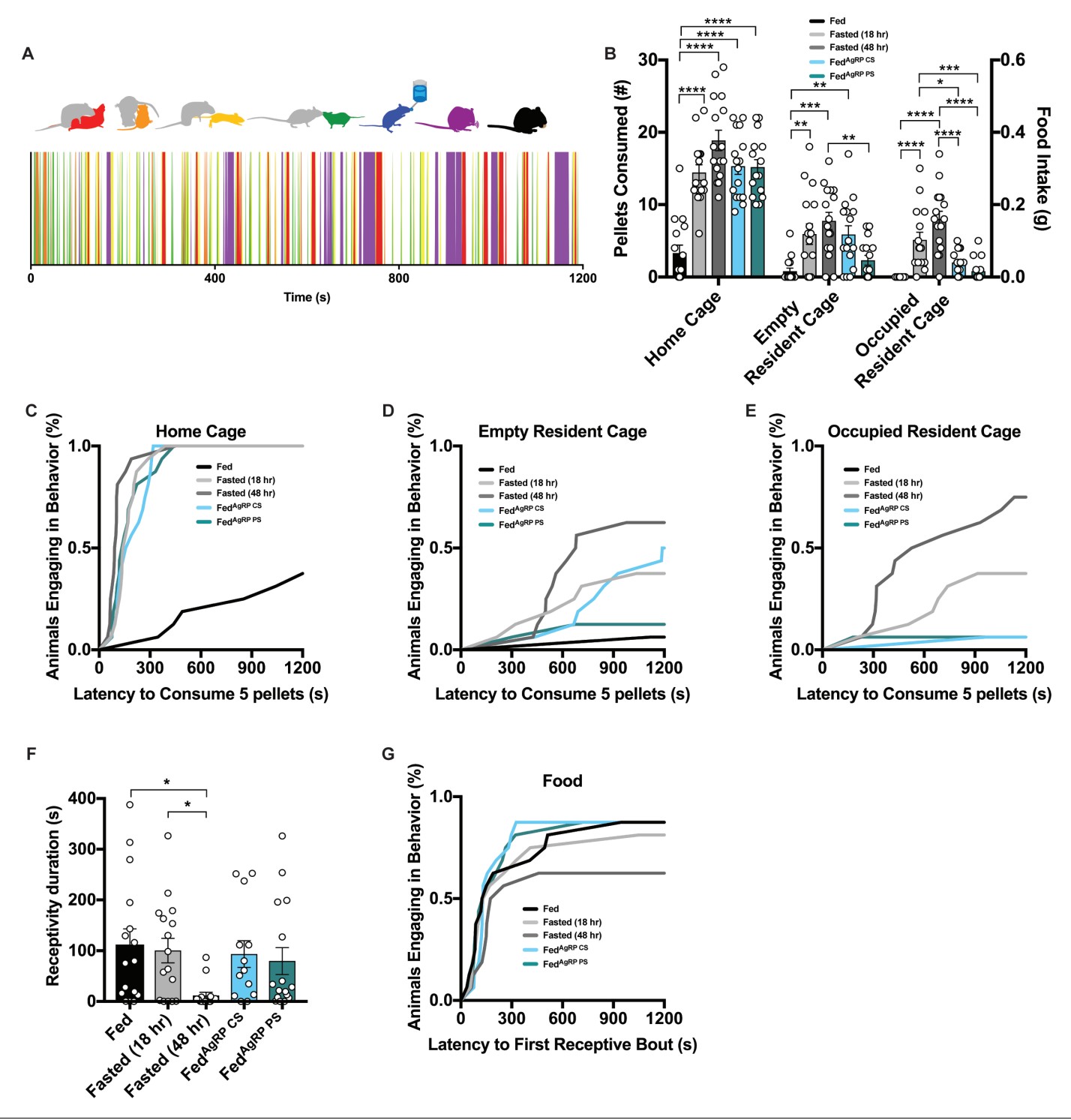

**Figure 4.** Evaluating female intruder behavior in response to a dominant male resident. (A) Catalogue of scored female intruder behaviors (top); receptivity, encompassing presentation/lordosis (red), non-receptivity, comprising of escape/attempted escape (yellow), anogenital chemoinvestigation (orange), nose-to-nose chemoinvestigation (green), drinking (blue), grooming (purple) and eating (black). Sample raster plot of second-to-second scored behaviors across the entire Trial Phase (bottom). (B–E) Quantified between-subject comparisons of female intruder food intake (B) and cumulative latency plot to consume 0.1 grams of food in the intruder homecage (C), empty resident cage (D) or occupied resident cage (E). (F–G) Quantified between-subject comparisons of female intruder receptivity behavior (F) and cumulative latency plot to first receptive bout (G) in the presence of food. n = 16 per group, values are means ± SEM. *p<0.05, **p<0.01, ***p<0.001.
*Figure 4 continued on next page*

*Figure 4 continued*

DOI: https://doi.org/10.7554/eLife.44527.009
The following figure supplements are available for figure 4:
**Figure supplement 1.** Experimental design of resident-intruder assay during evaluation of intruder behavior.
DOI: https://doi.org/10.7554/eLife.44527.010
**Figure supplement 2.** Evaluating female intruder behavior in response to a dominant male resident.
DOI: https://doi.org/10.7554/eLife.44527.011

aggress before trials were conducted to gain fighting experience and establish dominance in their home cage, as a history of winning previous territorial encounters increases the probability of aggressive behaviors (*Barnett and Spencer, 1951*; *Calhoun, 1963*; *Crowcroft, 1955*; *Crowcroft, 1966*; *Crowcroft and Rowe, 1963*).

After this adaptation phase, male resident mice were tested in each of the five hunger states (*Figure 2—figure supplement 1A*). Identical to above, the resident male-intruder assay was comprised of 3 phases (Pre, Trial and Post) and each animal was tested in three distinct Trial-Phases (*Figure 2—figure supplement 1D*, middle panel). *Ad libitum* fed, group-housed males ranging from 5 to 8 weeks old and 15–22 grams without fighting experience were selected to serve as intruders (*Figure 2—figure supplement 1C*) as subordination is highly dependent on housing status, age, size and fighting experience (*Crowcroft, 1955*; *Crowcroft, 1966*; *Crowcroft and Rowe, 1963*; *Yang et al., 2017*).

## Aggressive encounters do not impact feeding, but hunger impacts aggressive behaviors

Resident behavior in the male intruder assay was scored in a similar manner to that of the resident-female intruder paradigm above. Interactive behaviors comprised of aggressive bouts of attack (red), chasing (yellow), anogenital chemoinvestigation (orange) and nose-to-nose chemoinvestigation (green), as well as individual behaviors encompassing drinking (blue), grooming (purple) and eating (black) were recorded on a second-to-second basis across the 20 min trial (*Figure 5A*, top). Collectively, this selection of behaviors was then used to generate raster plot sequences encoding each resident animal's behavior over the course of the experimental trial phase (*Figure 5A*, bottom; *Figure 5—figure supplement 1A*).

Food intake measurements in the male residents in the absence of a male intruder affirmed both physiological and artificially-induced hunger stimulate feeding coincident on need (*Figure 5B*). Reflective of this consummatory behavior, the latency to eat 0.1 grams (five pellets) of chow was comparable across the natural and artificial hunger states with each mouse meeting this threshold (*Figure 5C*). The same mice in the Fed condition ate significantly less and only a small minority reached the threshold of consuming five pellets (*Figure 5B–C*).

As in the resident-female intruder paradigm above, a similar pattern of food intake was present contingent on caloric need state in the presence of a male intruder (*Figure 5B*). However, unlike our findings in the resident-female intruder paradigm (*Figure 2—figure supplement 2B–F*), when a male intruder was introduced into the resident's cage, the total amount of food consumed by the resident mice did not vary much from the no intruder condition (*Figure 5B*; *Figure 5—figure supplement 1B–F*). These results were echoed by the cumulative latency to consume 0.1 grams of food, as latency curves remained comparable in the presence or absence of a male intruder (*Figure 5C–D*; *Figure 5—figure supplement 1G–K*). Therefore, unlike a female intruder, a subordinate male stimulus was insufficient to curtail total food consumption across appetite states.

In the Post-Phase after the intruder male was removed from the cage, all groups of resident animals, regardless of hunger state, consumed more food compared to the same period following a Trial Phase in the absence of a male intruder (*Figure 5—figure supplement 2A–F*).

Of all the conditions tested, Fasted (48 hr) and Fed$^{AgRP\ CS}$ conditions displayed the lowest level of territorial aggression, reflected by the total duration of time (*Figure 5H*), bout number (data not shown) and total number of residents aggressing intruder mice (*Figure 5I*). Although the latency to attack is quick, the speed at which the residents aggress the intruder is slightly longer in conditions of hunger (*Figure 5I*). Owing to the variation and short duration of aggressive bouts, a regression

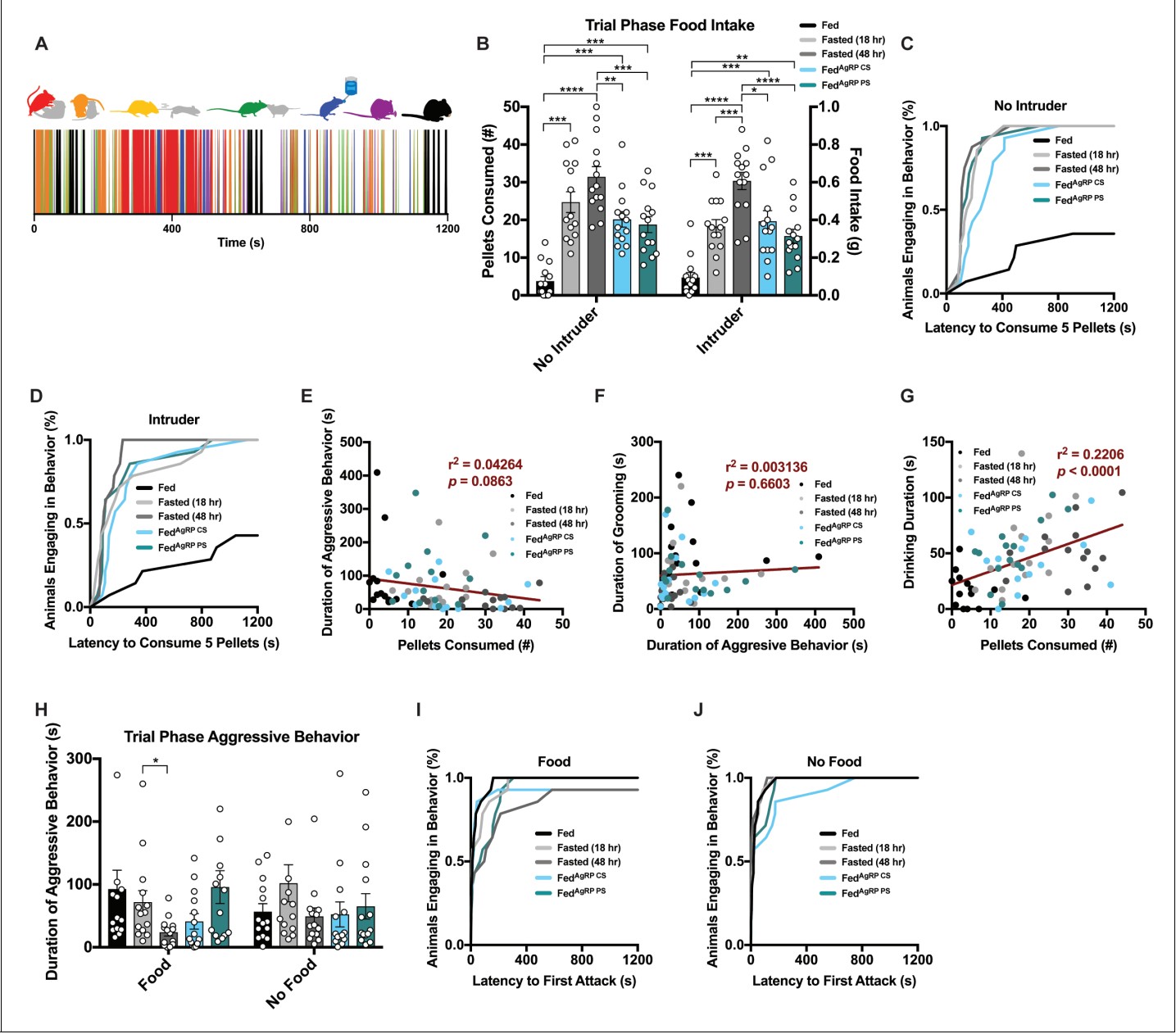

**Figure 5.** Evaluating resident behavior in response to a subordinate male intruder. (**A**) Catalogue of scored resident behaviors (top); aggressive bouts of attack (red), chasing (yellow), anogenital chemoinvestigation (orange), nose-to-nose chemoinvestigation (green), drinking (blue), grooming (purple) and eating (black). Sample raster plot of second-to-second scored behaviors across the entire Trial Phase (bottom). (**B–D**) Quantified between-subject comparisons of resident food intake (**B**) and cumulative latency plot to consume 0.1 grams of food in the absence (**C**) or presence of a male intruder (**D**). (**E–G**) Linear regression plots between aggression and feeding (**E**), grooming and aggression (**F**) and drinking and feeding (**G**). (**H–J**) Quantified between-subject comparisons of resident aggression behavior (**H**) and cumulative latency plot to first attack in the presence (**I**) or absence of food (**J**). n = 16 per group, values are means ± SEM. *p<0.05, **p<0.01, ***p<0.001.

DOI: https://doi.org/10.7554/eLife.44527.012

The following figure supplements are available for figure 5:

**Figure supplement 1.** Evaluating resident behavior in response to a subordinate male intruder.
DOI: https://doi.org/10.7554/eLife.44527.013

**Figure supplement 2.** Post-trial phase food intake in resident mice in male intruder paradigm.
DOI: https://doi.org/10.7554/eLife.44527.014

could not capture a relationship between fighting and feeding but the trend suggests a correlation exists (*Figure 5E*). While male mice often engage in post-coital grooming behavior following an encounter with a receptive female, they fail to show post-fight grooming after engagement with an intruding male as the dominant animal is often uninjured during the confrontation (*Barnett and Spencer, 1951*; *Crowcroft, 1955*; *Crowcroft and Rowe, 1963*). Concordantly, we did not detect any relationship between the duration of grooming with duration of aggressive behavior (*Figure 5F*). However, the residents in the male intruder assay demonstrated a positive correlation between drinking duration and food consumed (*Figure 5G*).

Aggression behavioral architecture was more similar across appetite states in the absence of food where animals exhibited comparable amounts of aggressive behaviors toward a male intruder (*Figure 5H*). When feeding was unattainable, latency to aggress male intruders occurred faster and every subject tested demonstrated aggressive behaviors toward male intruders when compared to the same residents in the presence of food (*Figure 5I–J*; *Figure 5—figure supplement 1Q–U*). Fasted (48 hr) animals significantly escalated the duration spent aggressing a male intruder in the absence versus presence of food, while the degrees of aggression expressed in all other groups were comparable independent of food accessibility (*Figure 5H*; *Figure 5—figure supplement 1L–P*).

### Satiating substrate choice substantially impacts behavioral sequences in resident males exposed to male intruders

To further investigate relationships between behavioral choices, we again analyzed action sequences as each resident animal transitioned from one behavior into another in the presence of (*Figure 6A*) or absence of food (*Figure 6B*). Using the Fed condition as a baseline, we found that specific patterns of behavior emerged with greater probability such as the transition from chasing into aggression, anogenital chemoinvestigation into aggression, nose-to-nose chemoinvestigation into aggression, aggression into aggression and grooming into grooming, while others occurred infrequently such as the transition from eating to eating, eating to grooming and drinking to drinking (*Figure 6A*, far left). We uncovered disruptions in these behavioral sequences in the same animals under distinct hunger conditions. For instance, the probability to transition from chasing into aggression or aggression into chasing decreased in escalating conditions of hunger, while the probability to transition from eating to eating increased in all hunger states regardless of how hunger was evoked (*Figure 6A*).

Using the cosine distance measure, we found the overall pattern of sequential behaviors was the most divergent from the Fed state in the Fasted (48 hr) and Fed$^{AgRP\ CS}$ conditions (*Figure 6C*). Similar to our observation in the female intruder assay, when we evaluated these sequences of behavior in the absence of food, transitions between behaviors across all five conditions were far more alike (*Figure 6B*). Accordingly, the cosine distances from the Fed state were comparable across all natural and artificial hunger states in the absence of food (*Figure 6C*). As above, the differences from the Fed state in our resident male-intruder assay were lower in the absence of food (*Figure 6D–G*), supporting the hypothesis that food availability and subsequent consumption is a major driver of these alterations in behavioral sequencing patterns.

## Subordinate male intruders display context- and state-dependent feeding

### Male intruders display environmental Context-Dependent food consumption

Next, we aimed to understand how varying degrees of physiological and artificially-evoked hunger govern a male intruder's behavior in a dominant resident's marked territory. Additionally, we were interested in examining the role of environmental context on feeding behavior. Cohorts of mice were prepared and tested as described above (*Figure 4—figure supplement 1A–B,D*). All male intruders were housed in grouped cages and were both younger and weighed less than the dominant residents.

A similar within-subject strategy was employed and intruder males were tested in three contexts (*Figure 4—figure supplement 1*, middle panel). As expected, we observed home cage food intake driven by caloric need, as all hungry animals, regardless of evoked-state, exhibited significantly

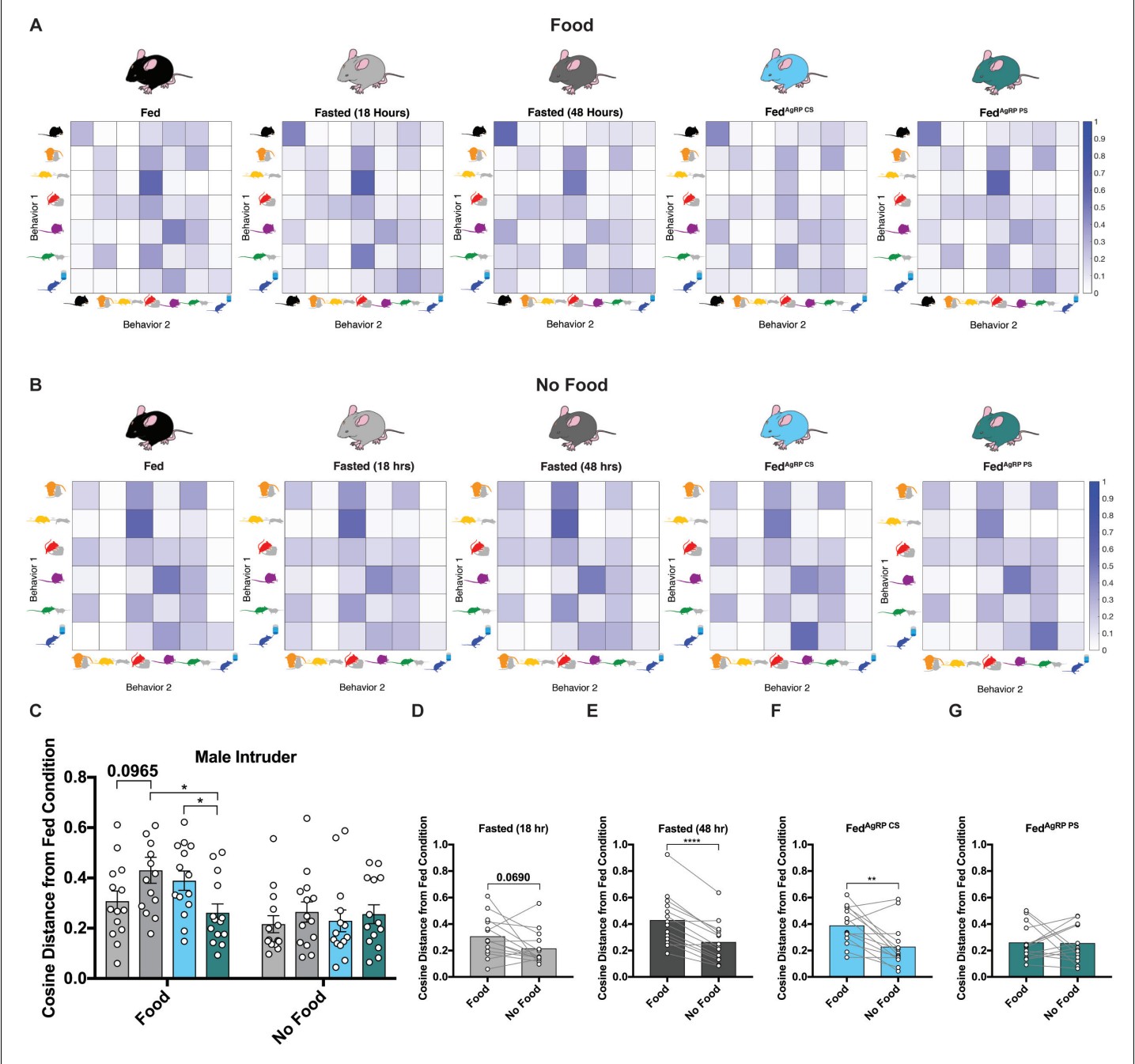

**Figure 6.** Behavioral sequence analyses of resident males exposed to a male intruder. (A–B) Transition matrices of behavioral transitions grouped by appetite condition in the presence (A) or absence (B) of food. (C) Quantified between-subject comparisons of the cosine distance from the Fed condition in the presence or absence of food, respectively. (D–G) Quantified within-subject comparisons of the cosine distance from the Fed condition in the presence versus absence of food. n = 16 per group, values are means ± SEM. *p<0.05, **p<0.01, ***p<0.001.
DOI: https://doi.org/10.7554/eLife.44527.015

increased food intake compared to Fed intruders with a short latency to procure five pellets (*Figure 7B–C*). Placing these same male intruder mice into an empty, but territorialized, resident's cage strongly suppressed feeding in all animals independent of hunger condition (*Figure 7B*; *Figure 7—figure supplement 1A,C,E,G,I*). Only a few hungry animals reached the 0.1 gram food intake threshold (*Figure 7D*; *Figure 7—figure supplement 1B,D,F,H,J*). This precipitous drop in feeding was further diminished in the territorialized, occupied male resident cage (*Figure 7B*; *Figure 7—*

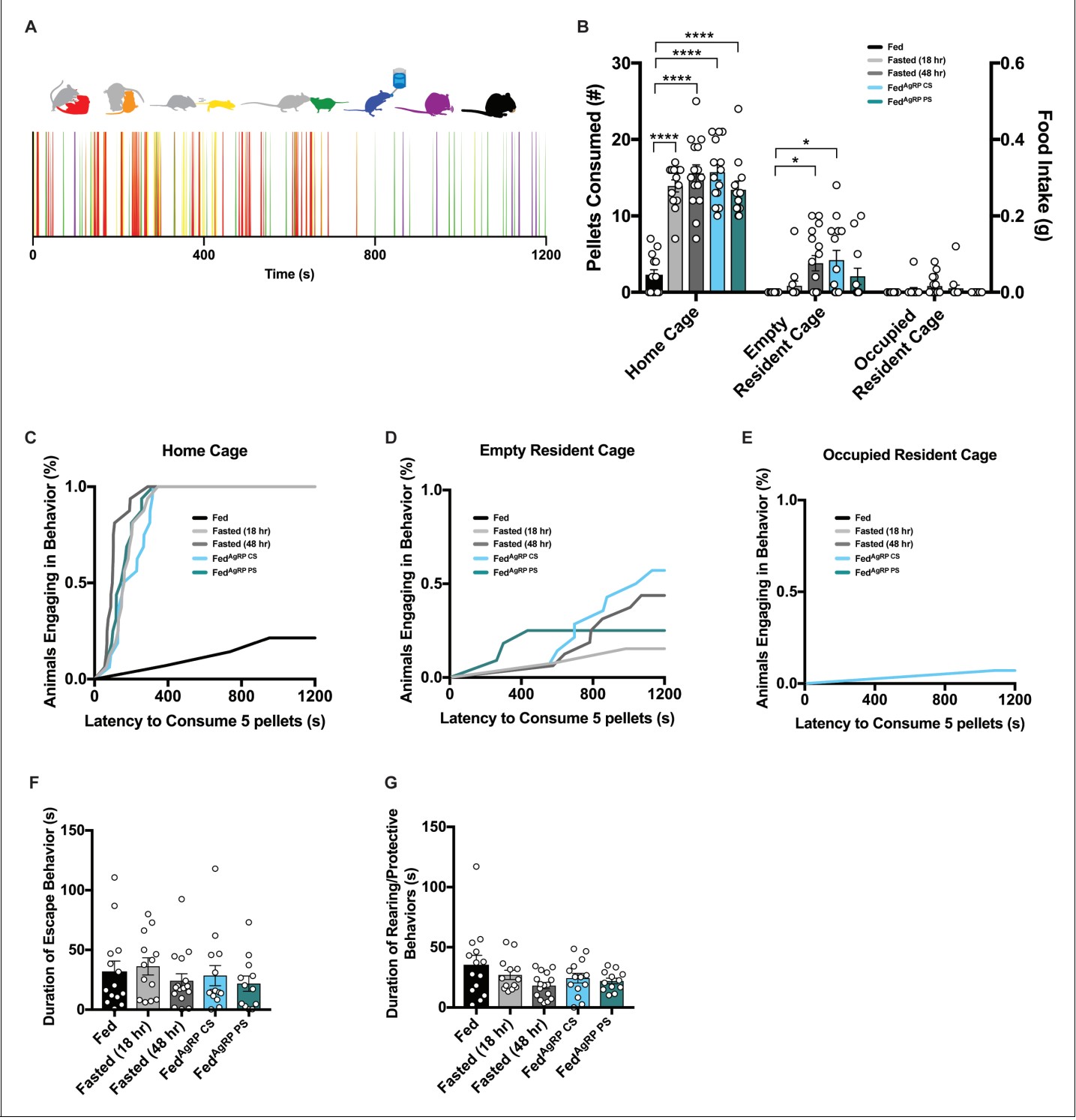

**Figure 7.** Evaluating male intruder behavior in response to a dominant male resident. (**A**) Catalogue of scored male intruder behaviors (top); defensive posturing such as rearing and being pinned (red), flight comprising of escape/jumping (yellow), anogenital chemoinvestigation (orange), nose-to-nose chemoinvestigation (green), drinking (blue), grooming (purple) and eating (black). Sample raster plot of second-to-second scored behaviors across the entire Trial Phase (bottom). (**B–E**) Quantified between-subject comparisons of male intruder food intake (**B**) and cumulative latency plot to consume 0.1 grams of food in the intruder homecage (**C**), empty resident cage (**D**) or occupied resident cage (**E**). Quantified between-subject comparisons of male intruder escape (**F**) and defensive (**G**) behaviors. n = 14 Fed group, n = 13 Fasted (18 hr) group, n = 16 Fasted (48 hr) group, n = 14 Fed^AgRP CS group, n = 12 Fed^AgRP PS group. values are means ± SEM. *p<0.05, **p<0.01, ***p<0.001.

DOI: https://doi.org/10.7554/eLife.44527.016

*Figure 7 continued on next page*

*Figure 7 continued*

The following figure supplement is available for figure 7:

**Figure supplement 1.** Evaluating male intruder behavior in response to a dominant male resident.
DOI: https://doi.org/10.7554/eLife.44527.017

*figure supplement 1A,C,E,G,I*). Only one animal of all the intruder males tested in this context reached the 5-pellet threshold (*Figure 7E*; *Figure 7—figure supplement 1B,D,F,H,J*). However, some of the male intruders initiated feeding during the Post-Phase following the removal of the male resident from the territorialized cage (Figure *Figure 7—figure supplement 1K*).

## Hunger state does not alter male intruder behaviors

A catalogue of 7 male intruder behaviors were scored throughout each Trial-Phase: both interactive behaviors including defensive posturing such as rearing and being pinned (red), flight comprising of escape/jumping (yellow), anogenital chemoinvestigation (orange) and nose-to-nose chemoinvestigation (green) and individual behaviors comprising drinking (blue), grooming (purple) and eating (black) were assessed (*Figure 7A*, top). These behaviors were scored across the 20 min Trial-Phase and used to create a behavioral barcode for each subject (*Figure 7A*, bottom). Unlike our observations in female intruders, we found all male intruder animals across groups demonstrated comparable behaviors in all metrics measured here. Male intruders were attacked quickly and displayed similar durations of flight/escape and defensive posturing (*Figure 7F–G*). As expected, the latency in which male intruders were forced to fend off attacks was short and showed no discretion based on hunger levels (data not shown).

## Discussion

Motivated behaviors, while of great interest to the field of neuroscience, are often studied in isolation to assess labeled lines of neural connections underlying the satiation of innate motives. In natural settings, however, multiple motivations must compete in real time for expression through goal-directed behaviors. As some motivations take precedence over others, animals must adjust their behavior to optimize survival on a moment-to-moment basis. Here, we demonstrate this phenomenon using hunger and intraspecific social interactions as two orthogonal motivations requiring animals to entertain one or both motives over time. We demonstrate that as caloric need rises, animals shift their behavior away from social interaction with female or male conspecifics in favor of feeding (*Figure 8A-C*). We also show that optogenetically driving ARC$^{AgRP}$ activity in calorically-replete animals during these interactions deters residents from social interactions with either sex, again in favor of feeding. However, the protocol used to evoke ARC$^{AgRP}$-mediated hunger (concurrent- versus pre-stimulation) leads to distinctive changes in behavioral architecture (see Discussion below).

### Strengthening motivation through experience

A systems model of motivational theory proposed by Toates and Singer describes the joint interaction between internal need and external incentives, highlighting the importance of learned information regarding previous associations with environmental stimuli (*Toates, 1980*; *Toates, 1981*). For example, we are more likely to consume food not only when our energy stores are low and food becomes accessible, but if our past encounters with food sources revive memories representing an innocuous relationship, or better, one of positive energy gain with no negative consequences. This relationship can be applied across motivational systems. For example, conspecifics of the opposite sex serve as incentive stimuli and arouse a motivation state to approach/mate; this drive intensifies with longer deprivation (period since the last ejaculation). Furthermore, a male's tendency to converge upon a female and copulate is stronger if such behavior in the past resulted in ejaculation (*Kagan, 1955*). Similarly, territorial aggression is amplified in dominant animals if attacks result in successful subordination of rival conspecifics (*Archer, 1976*). Thus, we potentiated responses of our experimental animals through experience, leading to learned representations regarding chow intake (restoration of calories), receptive female intrusion (copulation/ejaculation) and subordinate male intrusion (social dominance) before the assays were conducted.

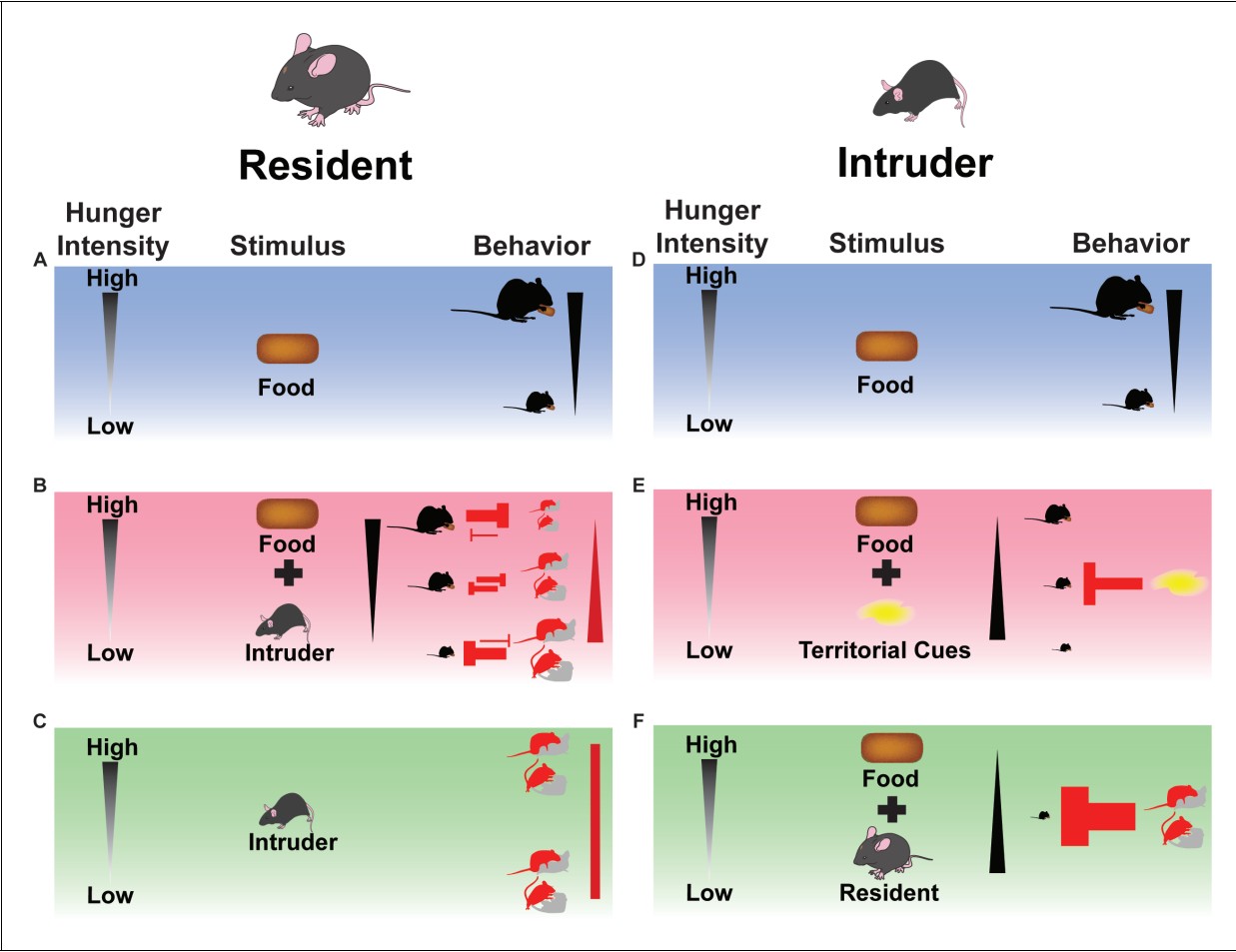

**Figure 8.** Summary Model: Feeding and social interactions/cues dynamically compete for behavioral expression. **(A)** When resident male mice are presented with a singular external incentive (food), they express feeding based on internal state (hunger intensity). **(B)** When resident male mice are presented with multiple external incentives (food and intruding animal), a conflict arises between the motivation to eat or mate/aggress. If hunger is minimal, social interaction drives will suppress eating behaviors. If hunger intensity is high, the drive to feed will in turn deprioritize social interaction behaviors. If need state is intermediate, behaviors will both be expressed in moderate amounts. **(C)** When resident male mice are presented with a singular external incentive (intruder), they express social interaction behaviors independent of internal state (hunger intensity). **(D)** When intruder mice are presented with a singular external incentive (food) in a familiar context (homecage), they express feeding based on internal state (hunger intensity). **(E)** When intruder mice are presented with multiple external incentives (food and territorial cues) in an unfamiliar context (empty resident cage), they attenuate feeding based on internal state (hunger intensity). **(F)** When intruder mice are presented with multiple external incentives (food and resident animal) in an unfamiliar context (occupied resident cage), they abrogate feeding independent of internal state (hunger intensity), deferring to the resident's behavioral choices of mating or aggressive advances.
DOI: https://doi.org/10.7554/eLife.44527.018

### Dichotomy between resident interactions with female versus male intruders

With respect to feeding, we observed an interesting division between resident interactions toward an intruding receptive female or subordinate male: while residents suppressed feeding upon exposure to a female intruder in all states investigated, they did not, for the most part, upon exposure to a male. It is possible female intrusion in general induces more robust arousal in the resident which could potentially displace food intake. More likely, the time spent engaging in stereotyped, sex-specific social behavior underlies this difference. Female-directed behaviors took quantitatively longer to evolve: total amount of time residents spent in pursuit of and mating with a female were lengthier than time spent aggressing an intruding male (compare *Figures 2H*,*5H*). It is possible that the amount of time remaining in the male intruder paradigm permitted the resident to eat comparable quantities of food as the same animals in the absence of a male intruder.

Furthermore, satiation of these needs culminate in disparate ways: motivation to reproduce is achieved by ejaculation, while motivation to aggress a male intruder ends in social defeat of the intruder. While these satiation points occur on highly variable timescales based on individual animal propensities for either motivation, actions leading to intromission/ejaculation are often more time-consuming than the establishment of dominance in a territorialized domain. After either of these points, interaction with either sex is disincentivized (*Collias, 1944*; *McGill, 1962*), allowing residents to engage in other behaviors throughout the rest of the trial (such as feeding). Thus, it is possible these differences result from the sheer limitation of the trial period we used.

## Variation in the sensitivity of mating and aggression to hunger state depends on food accessibility

Just as social behaviors shape feeding responses, hunger shows a profound effect on social interactions (*Figure 8B*). In concert with previous findings, we observed a robust decrease in the motivation to engage in reproductive or aggressive behaviors in animals subjected to prolonged food restriction (*Inaba et al., 2016*; *Padilla et al., 2016*). We demonstrate that ramping intensity of hunger states has scaled effects on metabolic endocrinology, which may in turn drive an exclusive set of behaviors. Past research indicates that hormones governing sexual and aggressive behaviors drop in calorically-deprived states (*Nikol'skaya et al., 2016*), leading to decreased reproductive behaviors in male mice (*Burns-Cusato et al., 2004*; *Shah and Nyby, 2010*), rats (*Jones and Wade, 2002*), and other rodents (*Li et al., 1994*), as well as diminished aggression toward other males (*Padilla et al., 2016*). Despite the commonly used term 'hangry' to describe a state of agitation while hungry, aggression research is mixed on the effects of hunger in potentiating aggression. Earlier research cites hunger as a catalyst for increased aggression (*Keys, 1946*; *Rohles and Wilson, 1974*). Furthermore, it has been shown across species that levels of aggression are amplified in the presence of a depletable food source (*Armstrong, 1991*; *Gray et al., 2000*; *Harris, 2010*; *Lim et al., 2014*). However, we found that as caloric need escalated, the drive to mate or aggress intruders dwindled. This subdual of social interactions could be the result of energy conservation or alternatively, a shift away from social behavior toward food consumption.

Arguing for the latter, we found that suppression of mating and aggression behaviors in hungry resident animals was reversed when food was absent (Figure 8C). Thus, by eliminating the option to sate hunger through food consumption, resident animals were able to devote more time to pursuing attainable incentives such as reproductive and territorial-defensive behaviors. Importantly, the absence of food to hungry mice, even those receiving ARC$^{AgRP}$-photoactivation, failed to lock animals into foraging programs in exclusion of competing need states. This makes sense on the level of ARC$^{AgRP}$ neural activity, which is suppressed by caloric information (*Betley et al., 2015*; *Beutler et al., 2017*; *Chen et al., 2015*; *Mandelblat-Cerf et al., 2015*; *Su et al., 2017*) but may not exhibit a homogeneous response to intruder animals in the absence of food. This result underscores the need for more comprehensive in vivo imaging of these neurons – and their downstream terminal fields – during interactions in various contexts. Promising targets include the medial amygdala (MeA), a region previously implicated in ARC$^{AgRP}$ suppression of aggression and self-preservation behaviors (*Padilla et al., 2016*), as well the paraventricular thalamus (PVT), a downstream site where ARC$^{AgRP}$ activity is sufficient to drive feeding (*Betley et al., 2015*) and influence cue valence processing (*Livneh et al., 2017*). Conversely, elucidating the putative influence caloric need acts on the endogenous activity patterns of canonical cell-types underlying social interaction will enhance our understanding of the dynamic interaction between motivational states.

## Behavioral choice sequence disrupted by extreme natural hunger and artificial 'hunger' induction

Due to the richness of our data set, we also sought to address disparities in sequential behavioral choice that emerged in our residents across hunger states. Transition matrices reporting the evolution from each scored behavior to another scored behavior revealed that while all hunger states showed differences from the Fed state in the presence of food, the starkest distinction belonged to the Fed$^{AgRP\ CS}$ condition. This finding is largely due not only to the patterns and sequences of behaviors but the frequency in which they occur. For example, resident animals in the Fed$^{AgRP\ CS}$ condition displayed reduced (but still present) mating advances and aggressive behaviors and

engaged in greater food intake than other appetite states. This disruption in behavioral sequences was eliminated when food was inaccessible, further demonstrating the power that external incentive availability, combined with internal state, exert on the optimal expression of behavior.

## A deeper investigation into ARC<sup>AgRP</sup> photoactivation

Although remote stimulation of ARC<sup>AgRP</sup> neurons reliably evokes food intake comparable to fasted animals (*Aponte et al., 2011*; *Krashes et al., 2011*), the activity of this neural subset is inhibited in the presence of food (*Betley et al., 2015*; *Chen et al., 2015*; *Mandelblat-Cerf et al., 2015*). Thus, optogenetically locking animals into this artificial state fails to mimic endogenous ARC<sup>AgRP</sup> activity in the presence of food, and thus is not an accurate representation of real-time network dynamics and ensuing behavior. However, every study to date has shown that concurrent ARC<sup>AgRP</sup> activation recapitulates behaviors observed in physiological hunger (*Alhadeff et al., 2018*; *Burnett et al., 2016*; *Dietrich et al., 2015*; *Goldstein et al., 2018*; *Krashes et al., 2011*; *Li et al., 2019*; *Padilla et al., 2016*; *Steculorum et al., 2016*), albeit many of these were performed in the absence of food. It was recently described that priming ARC<sup>AgRP</sup> neurons, via prestimulation in the absence of food, drives subsequent food intake upon availability at comparable levels as physiological or concurrent ARC<sup>AgRP</sup> activation (*Chen et al., 2016*), presenting a closer model of the dynamics of ARC<sup>AgRP</sup> neurons during natural hunger.

Therefore we tested our mice under both optogenetic protocols, revealing key differences in ARC<sup>AgRP</sup>-evoked behavior, particularly when animals were forced to decide between feeding and social interactions. While animals in both conditions presented similar magnitudes of food intake with no intruder present, animals' real-time behavior during intruder trials varied wildly. Moreover, to the best of our knowledge, we uncovered the first instances when Fed<sup>AgRP CS</sup> failed to phenocopy the behavior observed in physiologically fasted animals: resident males in the Fed<sup>AgRP CS</sup> condition demonstrated lower levels of mating and aggression behaviors toward intruders in the presence of food, when ARC<sup>AgRP</sup> activity should be silenced. Furthermore, although feeding was restrained in the presence of a female intruder, intake was not suppressed to the same degree as in all other appetite states tested. Along these lines, we report that the pattern of behavioral sequences varied more in this Fed<sup>AgRP CS</sup> condition compared to all other states tested, an effect that was rescued in the absence of food when natural ARC<sup>AgRP</sup> activity would remain elevated. Finally, the amount of food consumed by intruders in an occupied resident cage differed between the Fasted (48 hr) and Fed<sup>AgRP CS</sup> conditions suggesting additional alterations in physiology may be necessary to promote food seeking apart from ARC<sup>AgRP</sup> activity under this specific context. Thus, caution must be applied to interpreting function through artificial stimulation strategies.

## Environmental context can alter intruder behavior regardless of how hunger is elicited

Our study also identified comparable, but different, effects of territorial context on the execution of feeding behaviors during varying intensities of hunger (*Figure 8D-F*). We observed a robust decrease in food intake upon entry into an unknown resident's territory in both female and male mice across all hunger states compared to baseline feeding responses recorded in their homecage; this decrease was exacerbated by the resident occupancy. Accounts of natural mouse behavior indicate that dominant animals patrol and aggressively defend marked territory in the face of intrusion (*Crowcroft, 1955*; *Crowcroft, 1966*). Moreover, cage switch experiments where animals are placed in a dirty cage containing soiled bedding of another conspecific elevates body temperature and locomotor activity (*Lee et al., 2004*; *Piñol et al., 2018*). Thus, abrogation of intruder food intake is likely driven primarily by stress-induced changes in physiology independent of sex. We suspect the stress of entering a resident's territory also overrides concurrent ARC<sup>AgRP</sup> stimulation as past work suggests the timing of ARC<sup>AgRP</sup> activation onset influences behaviors oriented towards threat avoidance (*Jikomes et al., 2016*).

We also detailed a dichotomy in intruder responses to resident interactions: in females subjected to intense fasting conditions (Fasted 48 hr), we observed marked reductions in receptivity to mating advances, while we saw no differences in male intruder defense or escape behaviors across hunger states. A similar reduction was not revealed in the Fed<sup>AgRP CS</sup> condition, further substantiating key differences among natural versus artificially-induced hunger states. We draw several conclusions

from this pattern of results. First, female intruders have more agency over an interaction with a male resident: they can choose to display or reject a male's advances (*Johansen et al., 2008*), while subordinate male intruders must endure resident attacks regardless of their resistance. Second, this effect is likely influenced by hormones: akin to male sex hormones, those regulating female reproductive behavior show well-documented decreases in calorically-deprived states (*Baranowska et al., 2001*; *Olofsson et al., 2009*; *Wade and Schneider, 1992*). Long starvation periods can also cause overall disruption of the estrous cycle (*Judd, 1998*) and reduced reproductive behavior (*Wang et al., 2006*), effects that can be simulated by chronic ARC$^{AgRP}$ activity (*Padilla et al., 2017*). Male intruder escape and defensive behaviors remain constant across all appetite states, suggesting that even calorically-deprived or ARC$^{AgRP}$-evoked animals will defend themselves when residents attack. It is clear both female- and male resident-intruder interactions are highly driven by the resident animals' behavior towards the intruder, regardless of intruder hunger condition.

Overall, we have demonstrated more dynamic interaction between two mutually exclusive motivations – hunger and social interaction – than has previously been appreciated in recent neuroscience research. We have shown that in rodents, the context of a social interaction, including territorial ownership and hunger state, can prominently guide how behavioral encounters will unfold. Subsequent research should further address the neurophysiology of homeostatic feeding circuitry both upstream and downstream of ARC$^{AgRP}$ neurons that may principally alter the interplay of these two orthogonal motivations and their behavioral expression in real time. Furthermore, future high-level modeling of decision-making will advance the field's understanding of computational choice behavior.

## Materials and methods

### Animals

*Agrp-ires-Cre* (*Tong et al., 2008*) mice (The Jackson Laboratory; Stock No: 012899) were generated and maintained as previously described. All mice were back-bred onto a C57BL/6J background for at least three generations. All animal care and experimental procedures were approved by the National Institute of Health Animal Care and Use Committee. Mice were housed at 22–24°C with a 12 hr light:12 hr dark cycle with standard mouse chow (Teklad F6 Rodent Diet 8664; 4.05 kcal g$^{-1}$, 3.3 kcal g$^{-1}$ metabolizable energy, 12.5% kcal from fat; Harlan Teklad) and water provided *ad libitum*, unless otherwise stated. All diets were provided as pellets. For all behavioral studies mice of both sexes between 5–18 weeks were used. All mice used for resident social interaction behavior assays were singly-housed for female intruder assays, or co-housed with a female conspecific for male intruder assays. All mice used for intruder social interaction behavior assays were group-housed until the start of the experiments.

### Tissue weights and blood measurements

C57BL/6J mice were age- and weight-matched into their respective groups. Animals were euthanized in the Fed, Fasted (18 hr) and Fasted (48 hr) state and tissue and blood was collected. Free fatty acids (FFA, Roche Diagnostics GmbH, Mannheim, Germany) and triglycerides (Pointe Scientific Inc, Canton, MI), were measured using the indicated colorimetric assays. D-Lactate and beta-hydroxybutyrate were measured using calorimetric assays (BioVision, Milpitas, CA). Leptin (R and D Systems, Minneapolis, MN) and insulin (Crystal Chem, Downers Grove, IL) were measured by ELISA. Corticosterone was measured by RIA (MP Biomedicals, Orangeburg, NY). Liver triglyceride was measured as glycerol after NaOH hydrolysis (BioVision, Milpitas, CA).

### Body composition analysis

Measures of fat and lean body mass were determined in live mice using quantitative magnetic resonance (QMR) spectroscopy (EchoMRI 3-in-1, Echo MRI). Daily measurements were taken across 3 days.

### Viral injections

Stereotaxic injections were performed as previously described (*Krashes et al., 2013*). Mice were anesthetized with isoflurane and placed into a stereotaxic apparatus (Stoelting Just for Mice). For

postoperative care, mice were injected intraperitoneally with meloxicam (0.5 mg per kg). After exposing the skull via small incision, a small hole was drilled for injection. A pulled-glass pipette with 20–40 mm tip diameter was inserted into the brain and virus was injected by an air pressure system. A micromanipulator (Grass Technologies, Model S48 Stimulator) was used to control injection speed at 25 nl min$^{-1}$ and the pipette was withdrawn 5 min after injection. AAV10-CAG-FLEX-ChR2(H134R)-tdTomato (University of Pennsylvania School of Medicine; titer $1.3 \times 10^{13}$ genome copies per ml) was unilaterally injected into the arcuate nucleus of the hypothalamus (ARC; 200–300 nl, bregma: AP: –1.44 mm, DV: −5.70 mm, ML: ±0.25 mm).

## Optic fiber implantation

Optical fibers (200 μm diameter core; BFH37-200 Multimode; NA 0.37; Thor Labs) were implanted unilaterally over the ARC (bregma: AP: −1.45 mm, DV: −5.45 mm, ML: −0.20 mm). Fibers were fixed to the skull using C and B Metabond Quick Cement and dental acrylic and mice were allowed 2 weeks for recovery before acclimatization investigator handling for 1 week before experiments. Following implantation, resident animals were singly-housed while intruder animals were put back into their grouped-cages.

## Photostimulation protocol

Fiber optic cables (1 m long, 200 mm diameter; Doric Lenses) were firmly attached to the implanted fiber optic cannulae with zirconia sleeves (Doric Lenses) and coupled to lasers via a fiber-optic rotary joint (Doric Lenses). During photostimulation experiments, light pulse trains (5 ms pulses of 20 Hz; 2 s on, 2 s off) were programmed using a waveform generator (PCGU100; Valleman Instruments or Arduino electronics platform) that provided input to a blue light laser (473 nm; Laserglow). We adjusted the light power of the laser such that the light power exiting the fiber optic cable was 10–12 mW using an online light transmission calculator for brain tissue http://web.stanford.edu/group/dlab/cgi-bin/graph/chart.php. We estimated the light power at the ARC at 4.99 mW/mm$^2$. Note that this is likely a high estimation because some light was probably lost at the interface between the fiber optic cable and the implanted fiber optic cannula.

## Screening protocols

Screening for all animals receiving ARC$^{AgRP}$ optogenetic stimulation was conducted at ZT2-3 (9-10AM; near the beginning of the light cycle when baseline food intake is low) and occurred after experimenter handling acclimation, at least 3 weeks after surgical procedures. Animals to be used for intruder trials were singly-housed overnight before screening to ensure accurate measurement of food intake. On the day of the protocol, animals were tethered to a fiber-optic cable and allowed to acclimate for 10–20 min. The animals were given *ad libitum* access to 20 mg chow pellets (TestDiet) during a 20 min baseline, after which the number of pellets eaten was recorded. Following this baseline period, photostimulation was initiated for 20 min, and food intake was again measured after at the conclusion of this period. Only animals that consumed >0.30 g food intake (singly-housed residents) or >0.20 g (group-housed residents moved temporarily to single-housing for screening) were used for further experimentation.

## Social interaction assay: Resident analyses

All trials probing social interaction were conducted in a PhenoTyper (Noldus, Inc) home cage apparatus. The apparatus contained an overhead camera for trial monitoring and an automated pellet-dispenser, or Feeding Experimentation Device (F.E.D., *Nguyen et al., 2016*), which dispensed 20 mg grain pellets (TestDiet). Resident animals only had access to standard chow pellets (Teklad F6 Rodent Diet 8664, see Animals above) and *ad libitum* water when trials were not in progress. All residents lived in the apparatus for at least two weeks before trials, allowing for sufficient time to mark the cage with urine and feces and establish territorial domain. Residents used for the female-intruder assay were co-housed overnight with a female intruder mouse no less than 3 times during this 2 week acclimation period to provide male residents with sexual experience. Importantly, residents used for the female-intruder assay were sexually-deprived >48 hr. Residents used for male-intruder assay were co-housed with a female before and after trials. Critically, residents used for the male-intruder assay were challenged with five novel, younger, smaller, group-housed male conspecific

intruders during this 2 week acclimation period to provide experience aggressing male intruder mice.

Animals were tested in all experiments at ZT4-5 (10-11AM). Trials proceeded in three 20 min phases: Pre-Phase, Trial-Phase, and Post-Phase. During the Pre-Phase, all residents were attached to a fiber-optic cable (even if they were not receiving or going to be receiving photostimulation) and allowed to acclimate to attachment to the fiber. Notably, food was no accessible during this period. Resident mice in the ARC$^{AgRP}$ prestimulation group (Fed$^{AgRP\ PS}$) received 20 min of photostimulation at this time.

While residents were being acclimated to the fiber-optic cable, intruders were selected. Intruders were all aged 6–14 weeks from group-housing environments; all animals of both sexes were virgins. To determine female intruder eligibility, vaginal cytology was assessed for estrous state (see Vaginal cytology below); only females in the estrous phase were used.

The 20 min Trial Phase took on three distinct variations. Resident mice were either provided 1) *ad libitum* access to the F.E.D. in the absence of an intruder 2) *ad libitum* access to the F.E.D. in the presence of an intruder or 3) no access to the F.E.D. in the presence of an intruder. Animals were allowed to interact uninterrupted for 20 min.

Finally, during the 20 min Post-Trial period, intruder animals were removed from the apparatus, weighed, and quickly euthanized via carbon-dioxide inhalation. Resident animals remained in their homecages and were given 20 min *ad libitum* access to the F.E.D.

Pellets taken from the F.E.D. were automatically registered in the software but were verified via video to determine which specific animal consumed the food. We used the latency to consume 0.1 grams or five pellets as a metric to determine homeostatic feeding based on pellet consumption during the sated condition. Importantly, standard ~4 gram chow pieces were provided to all animals between experiments. Thus, exposure to the 20 mg chow pellets is limited to the experimental assays. On average, residents consumed ~5 pellets in the Fed state with less than 50% of the cohort meeting this threshold, demonstrating that this consumption was not physiologically necessary given they had *ad lib* food availability up until the start of the trial. Therefore, any income above five pellets consumed in the Fed state is likely a result stemming from the intermittent access to the 20 mg food pellets.

## Social interaction assay: Intruder analyses

Intruder behavior analysis occurred in the same apparatus as resident behavior analyses (see above). Intruder feeding was tested in three different contexts: in the home cage, in a resident-territorialized PhenoTyper cage, and in a territorialized cage while the cage's resident is present.

For home cage feeding, intruders were separated into single-housing cages containing a ramekin dish for one dark cycle. At ZT4-5 (10-11AM), the mice were provided with *ad* libitum access to 20 mg chow pellets and food intake was assessed over a 20 min period.

For empty resident cage intake assessment, residents and their female cohabiters were moved to a holding cage until after the trial. Intruders were tethered to a fiber-optic and allowed to acclimate to experimental handling (Fed$^{AgRP\ PS}$ received prestimulation during this time). The animals were then introduced to the empty resident cage with *ad libitum* access to the F.E.D. and feeding was assessed after a 20 min period.

For the occupied resident cage intake assessment, intruders were introduced into the resident's cage and provided with *ad libitum* access to the F.E.D. in the presence of a dominant male resident. All female intruders tested positive for estrus state the morning they were used for the intruder behavior trials.

After a 20 min interaction period, the residents were removed, weighed, and moved to a holding cage until the end of the intruders' 20 min Post-Trial phase where the intruder was provided with *ad libitum* access to the F.E.D. in the absence of the dominant resident. Finally, after the Post-Trial phase, intruders were removed from the cages, weighed, and quickly euthanized via carbon-dioxide inhalation. Resident animals were returned to their cages.

## Behavioral analyses

All trials were video recorded with a ceiling-mounted camera and used to score behaviors. All mice were scored for three individual behaviors: drinking, grooming and eating. Each mouse was also

scored for four interactive behaviors: nose-to-nose chemoinvestigation and anogenital chemoinvestigation independent of sex or resident/intruder status. Resident males were scored for attempted mating/pursuit and mounting/intromission in the presence of a female intruder and chasing and aggressive attack behavior in the presence of a male intruder. Female intruders were scored for escape/attempted escape and sexual receptivity, while male intruders were scored for flight/escape and defensive postures including rearing and being attacked. All behaviors were manually scored using JWatcher 1.0 (Daniel T Blumenstein, Janice C Daniel, and Christopher S Evans, http://www.jwatcher.ucla.edu/). Data was collected every second across the 20 min trial and if the animal performed one of these scored behaviors it was time-stamped and color-coded appropriately. Raster plots or behavioral barcodes were generated for each animal to provide a raw data snapshot of the animals sequence of behavior during the trials. Of note, we only scored discernable, easily-identifiable behaviors to verify reproducibility. Therefore, if the animals were performing other behaviors separate from our designated repertoire of actions, those particular actions went unaccounted for in our analyses. Importantly, the exact moment an animal took a pellet from the F.E.D. was automatically recorded, however video footage was used to (1) confirm the animal ate the pellet and (2) determine which animal retrieved and consumed the pellet in the case when multiple animals were occupying the cage.

Inter-rater reliability measures were conducted using confusion matrices directly comparing person-to-person scoring methods. Similarity of overall behavior scoring averaged greater than 70% coherence, as has been previously cited (Jhuang et al., 2010), and overall durations of behaviors scored show no significant differences between scorers.

## Sequence analysis

For each individual subject, in each behavioral state, sequences coded in JWatcher (see Behavioral Analyses) were extracted using custom Python scripts (Ilona Szczot, Science Exchange, Inc; Source code 1). Sequences were analyzed time-agnostically, meaning that inter-behavioral intervals and total duration of individual behavior bouts were not considered for this analysis. Using MATLAB (Source code 2), we analyzed the normalized frequency of transitions from each behavior to the following behavior for each animal as follows. First, the total number of bouts was normalized within each animal, thus removing inter-animal variability in number of behavioral bouts. A frequency proportion was calculated for each behavioral transition pair. Values close to 0 represented transitions that were infrequent or absent, and values near 1.0 represented sequences that occurred with high frequency. Within each feeding state, these transition frequency values were then arranged into multidimensional vectors. For each subject, we calculated the distance between the vector representing each individual's Fed state transition frequency and their natural or artificial hunger transition averages (producing one cosine distance calculation for each hunger state), thus providing within-subject data to be compared across conditions. To calculate this distance, we used the cosine method because it takes into account the overall pattern of averaged transitions across animals, constrains the average values between 0 and 1, and allows for more sensitive detection of differences than the more standard Euclidean distance measurement. Resultant cosine distance values were then compared using standard hypothesis-testing statistical analysis in GraphPad Prism 7.0 (see Statistical Analysis).

Transition matrices represent the average of transitions across animals within individual hunger states calculated as described above. Heat maps displayed in Figures 3 and 6 display across subject averages of each individual frequency, with lighter colors showing lesser frequency and darker colors greater frequency.

## Vaginal cytology

Sample collection and estrous cycle determination protocols were adapted from previous methods (Byers et al., 2012). Briefly, we inserted a pipette tip 1 mm inside the vagina of restrained female mice, then gently flushed 10 μL of 0.9% saline inside. The saline was then deposited onto a microscope slide (Fisher Scientific, Inc) and observed under a light microscope for cell contents in the saline drop before the drop was allowed to fully dry. We determined mice were in the estrous phase if almost all cells observed were cornified epithelial cells.

To stain cells for visualization, slides were allowed to fully dry before beginning staining. Slides were thoroughly rinsed in 1X phosphatase-buffered saline (PBS), then cells were fixed in 4% paraformaldehyde solution for 30 min. After fixing, slides were washed thoroughly in deionized water. Cells were stained in a 0.1% Toluidine Blue O solution (1% Toluidine Blue O in 70% ethanol, diluted to working concentration with 0.9% saline) for 3 min, then thoroughly rinsed in deionized water. Slides were then rapidly dehydrated in ramping in a step of 90% ethanol solution, then two steps of 100% ethanol. Samples were then cleared in two steps of 100% xylene for 3 min each step. Slides were then covered and imaged under light microscope.

## Statistical analysis

Statistical analyses were performed using Prism 7.0 (GraphPad) software. Data were extracted from JWatcher analysis files using original Python scripts/code written for analyses (Ilona Szczot, Science Exchange, Inc). Transition analyses were conducted in MATLAB (R2018a) using a combination of scripts (Theresa Desrochers; Brown University) and original scripts. In all statistical tests normal distribution and equal variance was established. The data presented met the assumptions of the statistical test employed.

## Acknowledgments

We thank Brad B Lowell for providing *Agrp*-ires-Cre mice used in this study. We thank the NIDDK Metabolic Core (directed by Oksana Gavrilova) for serum bloodwork and tissue weight analyses included in this study. We thank Ilona Szczot (Science Exchange) for original Python code used for data extraction from raw data files. We thank Alexxai V Kravitz and Katrina P Nguyen for guidance on assembling the Feeding Experimentation Devices for this study.

## Additional information

### Funding

| Funder | Grant reference number | Author |
| --- | --- | --- |
| National Institutes of Health | 1ZIADK075087-04 | Michael J Krashes |
| National Institutes of Health | 1ZIADK075090-04 | Michael J Krashes |

The funders had no role in study design, data collection and interpretation, or the decision to submit the work for publication.

### Author contributions

C Joseph Burnett, Conceptualization, Supervision, Visualization, Writing—original draft, Writing—review and editing; Samuel C Funderburk, Conceptualization, Data curation, Formal analysis, Investigation, Methodology, Writing—original draft, Writing—review and editing; Jovana Navarrete, Conceptualization, Data curation, Formal analysis, Investigation, Methodology, Writing—review and editing; Alexander Sabol, Data curation, Formal analysis, Investigation, Methodology, Writing—review and editing; Jing Liang-Guallpa, Data curation, Investigation; Theresa M Desrochers, Data curation, Formal analysis; Michael J Krashes, Resources, Software, Methodology

### Author ORCIDs

Michael J Krashes ⓘ http://orcid.org/0000-0003-0966-3401

### Ethics

Animal experimentation: This study was performed in strict accordance with the recommendations in the Guide for the Care and Use of Laboratory Animals of the National Institutes of Health. All of the animals were handled according to approved institutional animal care and use committee (IACUC) protocols (Protocol #K011-DEOB-16) of the NIH. The protocol was approved by the Committee on the Ethics of Animal Experiments of the NIH (Protocol #K011-DEOB-16). All surgery was performed under isoflurane anesthesia, and every effort was made to minimize suffering.

Decision letter and Author response
Decision letter https://doi.org/10.7554/eLife.44527.023
Author response https://doi.org/10.7554/eLife.44527.024

## Additional files

### Supplementary files
• Source code 1. Custom python scripts.
DOI: https://doi.org/10.7554/eLife.44527.019

• Source code 2. Custom MATLAB script.
DOI: https://doi.org/10.7554/eLife.44527.020

• Transparent reporting form
DOI: https://doi.org/10.7554/eLife.44527.026

### Data availability
All data generated or analyzed during this study are included in the manuscript and supporting files.

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
