## [Decision Letter]

Thank you for submitting your article "Need-Based Prioritization of Behavior" for consideration by *eLife*. Your article has been reviewed by three peer reviewers, one of whom is a member of our Board of Reviewing Editors, and the evaluation has been overseen by Catherine Dulac as the Senior Editor. The following individuals involved in review of your submission have agreed to reveal their identity: Garret D Stuber (Reviewer #2).

The reviewers have discussed the reviews with one another and the Reviewing Editor has drafted this decision to help you prepare a revised submission.

Summary:

Burnett and colleagues describe results from a series of studies aimed at identifying the physiological interaction of circuits controlling hunger and social interactions focusing on AgRP neurons. They used multidisciplinary approaches including behavior and optogenetics. The authors performed detailed behavioral analysis to determine how satiety states alter the expression of ethologically relevant social behaviors such as mating and territorial aggression. The major finding of the paper is that different degrees of food deprivation dramatically alter behavioral patterns within mice when competing motivationally relevant stimuli are present in the environment. Collectively, the studies are well described and provide novel observations that will be of wide interest. Several technical and presentation issues need to be addressed. Moreover, the flow and presentation of data is sometimes long and hard to follow. Potentially streamlining the results would make the article easier to digest for a larger audience.

Essential revisions:

The manuscript as written is difficult to digest. It seems that the manuscript has been written for a different journal format. For example, the Results section includes a lot of commentary that may be better suited for the Discussion. That being said the Discussion is a bit long.

The authors are to be commended for including all of the data. However, the figures are quite dense and routinely have panels from A-N. At a minimum, the individual panels are too small for my old eyes. The authors are encouraged to use the *eLife* figure format which allows supplements to figures that should allow some of the panels in a given figure to be supplements to that figure, so all of the data are there, but each part of the figure can be simpler and bigger.

For a journal with wide readership, a summary model figure would be helpful to convey the big picture to the readers.

The authors could expand the Discussion slightly to address how they think competing needs are processed at the neurocircuit level. Is there evidence from the literature that certain circuits become disengaged by changes in the hormonal state or under conditions of different types of deprivation?

---

## [Author Response]

Essential revisions:The manuscript as written is difficult to digest. It seems that the manuscript has been written for a different journal format. For example, the Results section includes a lot of commentary that may be better suited for the Discussion. That being said the Discussion is a bit long.

We thank the authors for the feedback and agree this study’s complexity has led the manuscript as initially written to be difficult to read. We have scaled back visual data presentation and restricted our interpretation of results and saved this for the Discussion/Materials and methods section to make it easier to read. In addition, we have also tightened up the Discussion section to address several of the main takeaways of the study and integrate reviewer suggestions.

The authors are to be commended for including all of the data. However, the figures are quite dense and routinely have panels from A-N. At a minimum, the individual panels are too small for my old eyes. The authors are encouraged to use the eLife figure format which allows supplements to figures that should allow some of the panels in a given figure to be supplements to that figure, so all of the data are there, but each part of the figure can be simpler and bigger.

We have altered our figures to facilitate easier interpretation of the Results section. In many instances, several graphs have been combined into one plot, inviting direct comparisons to be made more easily across conditions. Other pieces of main figures have also been moved to supplementary material. Furthermore, we removed the original Supplementary Figures 5 and 10 from the resubmission as we found these results redundant with the results as portrayed in the matrices of Figures 3 and 6, respectively. Similarly, data that was previously presented such as individual appetite state correlations or trial+post food intake measurements were removed due to the redundancy.

For a journal with wide readership, a summary model figure would be helpful to convey the big picture to the readers.

We appreciate this suggestion and have provided a summary model figure as a final main figure (Figure 8).

The authors could expand the Discussion slightly to address how they think competing needs are processed at the neurocircuit level. Is there evidence from the literature that certain circuits become disengaged by changes in the hormonal state or under conditions of different types of deprivation?

We have now added further discussion on the potential circuit mechanisms involved in behavioral choice and future experiments that can be done to begin to address these hypothesis. However, it is our opinion that this integration of binary choice behavior is extremely complex and will not occur at a singular loci in the brain. We have also emphasized previous literature that have explored the inhibition of competing behaviors such as pain perception, anxiety-related behavior, parental care, arousal, threat detection, compulsive behavior and social interactions by hunger intensity.

Although not discussed in detail due to space limitations, we would like to point out that hormones and circulating factors likely have dynamic effects regulating activity of neural circuits regulating feeding behaviors and social interactions. Ample evidence has investigated crossover effects of “conventional” hormone systems. Sex hormones such as testosterone and estrogen, for example, have dynamic, sex-dimorphic differences on energy metabolism (reviewed by Morford and Mauvais-Jarvis, 2016; Kelly and Jones, 2015); oxytocin also may act as an anorectic, as direct CNS infusion (Arletti et al., 1989) and intranasal treatments in humans (reviewed by Olszewski et al., 2017) can suppress food intake. In clinical populations, obesity’s interrelationship with other conditions is very clear: patients struggling with obesity exhibit a broad array of comorbid conditions, including low testosterone (Morford and Mauvais-Jarvis, 2016) and affective disorders (Luppino et al., 2010).

Much less is known, however, about specific locations at which these effects may occur, but we have some clues. The ARC, for example, densely expresses estrogen receptors (Esr1). Downstream regions of ARC^AgRP^ neurons, including the paraventricular hypothalamus (PVH) and bed nucleus of the stria terminalis (BNST), express oxytocin receptors and putatively play main roles in social interaction behaviors. Conversely, circulating factors signaling hunger or satiety such as leptin, insulin, and any number of gastrointestinal peptides likely act in multiple regions of the brain, especially in other nuclei of the hypothalamus.